# Quantitative analysis of relationship between mutation rate and speed of adaptation under antibiotic exposure in *Escherichia coli*

Atsushi Shibai[1]☉, Minako Izutsu[1,2,3,4]☉, Hazuki Kotani[1], Chikara Furusawa[1,5]*

1 Center for Biosystems Dynamics Research, RIKEN, Osaka, Japan, 2 Department of Microbiology, Genetics, and Immunology, Michigan State University, East Lansing, Michigan, United States of America, 3 BEACON Center for the Study of Evolution in Action, Michigan State University, East Lansing, Michigan, United States of America, 4 Ecology, Evolutionary Biology and Behavior Program, Michigan State University, East Lansing, Michigan, United States of America, 5 Universal Biology Institute, Graduate School of Science, The University of Tokyo, Tokyo, Japan

☉ These authors contributed equally to this work.
*chikara.furusawa@riken.jp

## Abstract

Mutations are the ultimate source of biological evolution that creates genetic variation in populations. Mutations can create new advantageous traits but can also potentially interfere with pre-existing organismal functions. Therefore, organisms may have evolved mutation rates to appropriate levels to maintain or improve their fitness. In this study, we aimed to experimentally quantify the relationship between the mutation rate and evolution of antibiotic resistance. We conducted an evolution experiment using 12 *Escherichia coli* mutator strains with increased mutation rates and five antibiotics. Our results demonstrated that the rate of adaptation generally increased with higher mutation rates, except in a single mutator strain with the highest mutation rate, which exhibited a significant decline in evolutionary speed. To further elucidate these findings, we developed a simple population dynamics model that successfully recapitulated the observed dependence of adaptation speed on mutation rate. These findings provide important insights into the evolution of mutation rate accompanied by the evolution.

## Author summary

Bacteria evolve to survive in changing environments, but how does the rate of genetic mutations influence their ability to adapt, especially to antibiotics? In this study, we explored how different mutation rates affect the speed at which *Escherichia coli* bacteria develop antibiotic resistance. By engineering bacterial strains with varying mutation rates and exposing them to five different antibiotics, we found that, in most cases, higher mutation rates led to faster adaptation. However, in one strain with an extremely high mutation rate, the ability to evolve was significantly reduced. Using a computational model, we confirmed this pattern and discovered that different antibiotics influence bacterial evolution in unique ways. Our findings help explain why some bacteria rapidly develop drug resistance while others struggle, providing valuable insights into antibiotic

**Data availability statement:** The resequencing analysis data have been deposited at the DDBJ Sequence Read Archive (https://www.ddbj.nig.ac.jp/dra/index-e.html) under accession number DRA016357 and DRA019544.

**Funding:** This work was supported by Japan Society for the Promotion of Science (JSPS; https://www.jsps.go.jp/english/) KAKENHI Grant Numbers 22K21344, 23K27164, 24H01798 (to C. F.), 19K16114 and 21K15077 (to A. S.), and Japan Science and Technology Agency (JST; https://www.jst.go.jp/EN/) ERATO Grant Number JPMJER1902 (to C. F.). The funder had no role in the study design, data collection and analysis, decision to publish, or preparation of the manuscript.

**Competing interests:** The authors have declared that no competing interests exist.

resistance. Understanding this balance between mutation and adaptation could lead to better strategies for controlling and preventing resistant infections.

## Introduction

Mutations have the ability to create new advantageous traits that are favored by natural selection, while having the potential to interfere with pre-existing organismal functions. Mutations exert a double-edged-sword influence on organisms, conferring either beneficial or deleterious effects [1], suggesting that mutation rates dynamically change over the course of evolution [1–3]. Specifically, under certain selection pressures, the speed of adaptive evolution can be accelerated by increased mutation rates, as evidenced in multiple studies [4–7]. These findings suggest that alleles modifying mutation rates can be subject to positive selection under certain conditions, or alternatively, they may become fixed in the population through hitchhiking on other beneficial mutations [4,8]. For instance, in a long-term experimental evolution study of asexual *Escherichia coli* populations, cells exhibiting significantly elevated mutation rates, termed 'mutators', have been observed [9,10]. In another example, an antibiotic selection experiment involved a mutator *E. coli* strain that initially exhibited an increased mutation rate due to the knockout of one gene. Subsequent increases in the mutation rate, caused by mutations in additional genes, were repeatedly observed [7]. However, when the mutation rate is excessively high, the overall fitness may decrease because deleterious mutations arise more frequently than beneficial mutations in most cases [2,10] and populations cannot avoid the accumulation of deleterious mutations. When a high mutation rate leads to a decrease in fitness, natural selection reduces the mutation rate. Indeed, several studies have demonstrated a reduction in the mutation rate during evolution [3,11,12]. The beneficial and harmful effects of mutations imply a complex and non-linear relationship between the mutation rate and the speed of fitness increase.

The experimental evolution of asexual bacterial populations aided by whole-genome sequencing is a powerful tool for investigating the effects of mutation rates on evolution [1,3–6]. Bacterial strains with different mutation rates can be prepared by deleting genes related to DNA replication or repair mechanisms as the mutator strains. Using these strains, we evaluated how changes in mutation rates affect the course of evolution. For example, Sprouffske et al. conducted experimental evolution using engineered *E. coli* strains with four different mutation rates [13]. These strains evolved for 3,000 generations in minimal medium without explicit stressors. Although populations with higher mutation rates had greater genetic diversity, this diversity benefited only when the mutation rate was modestly high. The study demonstrated that the highest mutation rates they used were not optimal for adaptation to the environment during long-term cultivation or for stress tolerance in novel environments after evolution.

Although the question of how evolutionary dynamics depend on mutation rates is important and has been the focus of many studies, the extent to which this relationship is influenced by a selective environment remains uncertain. This relationship is expected to depend on multiple factors including the distribution of fitness effects, population size, and the type of selection pressure. Analyzing the mutation rate dependency of evolutionary dynamics allows us to capture the contributions of these factors, providing a better understanding of how mutation rates evolve in nature and in laboratories. For example, the evolution of antibiotic resistance in microorganisms has been extensively studied in both laboratory and clinical settings [14–18] while the association between mutation rates and those events remains obscure. Given that recent studies have revealed drug resistance can be achieved by loss-of-function mutations [18] and studies

that show the fitness cost of resistant genotypes [19,20], it is likely that the distribution of fitness effects in environments that contain drugs is qualitatively different from that in environments without drugs, and hypermutator genotypes might be favored by selection in the former environments. Examining how evolutionary dynamics under antibiotic treatment depend on mutation rates will provide valuable insights into the broader mechanisms underlying the emergence of antibiotic resistance and the evolution of mutation rate itself.

In this study, we aimed to experimentally quantify the mutation-rate dependency of the evolution of antibiotic resistance. A recent study by Gifford et al. examined the advantage of a mutator sub-population for the evolution of drug resistance [21]. They changed the frequency of the mutator strain in the evolving populations while using the same mutator strain made by *mutS* deletion for all populations, which might cause biases on the mutation spectra [22]. To avoid capturing biased trends, we constructed 12 *E. coli* mutator strains with elevated mutation rates by deleting genes with different mutation spectra and combinations of those genes. It is important to note that the emergence of such multilocus mutators has been previously observed in antibiotic selection experiments [7]. Subsequently, starting with these mutator strains, we conducted evolution experiments using five different antibiotics, each with a distinct mechanism of action. The results revealed that the speed of adaptation, as measured by the rate of increase in the minimum inhibitory concentration (MIC), increased approximately linearly with mutation rate, except for a single mutator strain with the highest mutation rate, which exhibited a significant decrease in adaptation speed. Additionally, the results demonstrated that this dependency varied between bacteriostatic and bactericidal antibiotics. We successfully reproduced these mutation-rate dependencies using numerical simulations of a population dynamics model. These findings offer important insights into the evolution of mutation rates accompanying evolutionary processes.

## Results

### Construction of mutator strains

We used the *E. coli* MDS42 strain as the wild-type (WT) and generated knockout mutants of the *mutS, mutH, mutL, mutT*, and *dnaQ* genes, denoted as S, H, L, T, and Q, respectively. The *mutS, mutH*, and *mutL* genes are involved in the mismatch repair machinery [23], *mutT* plays a role in maintaining replication fidelity [24], and *dnaQ* codes for the epsilon subunit of DNA polymerase III [25]. Deletion of these genes causes a loss in replication fidelity and an increase in mutation rates. Additionally, we created seven double-gene-knockout strains and obtained 12 mutator strains, which are listed in Table 1. The sequence of gene deletions during the construction of the mutator strains is summarized in S1 Fig in S1 File.

To quantify the mutation rate, we conducted mutation accumulation (MA) experiments using mutator strains as ancestors. Specifically, we propagated three lineages for each ancestor as single colonies on an agar plate medium for 23-69 passages. We estimated the number of generations during the MA experiment by establishing a relationship between colony size and cell number [26–28]. Subsequently, we sequenced the samples at the end of the MA experiment to detect point mutations that had accumulated in the genome. The total number of identified base-pair substitutions (BPS) is summarized in Table 1.

We calculated the mutation rates per generation for each strain following a previous study [29]. Initially, we determined the proportion of each pattern of synonymous base-pair substitutions. Subsequently, we estimated the genome-wide mutation rate by dividing the number of accumulated synonymous mutations in a genome by the number of generations and then normalizing it with the frequency of possible mutational patterns. Additionally, the frequency of insertions and deletions (indels) was calculated.

**Table 1. Summary of MA experiments.** The two right-hand columns display the numbers of synonymous (Syn) and nonsynonymous (NSyn) base-pair substitutions (BPS) identified, respectively, while the mean and standard deviation of three replicates are presented. All strains except LQ had identical numbers of MA rounds among the three replicate MA lines, respectively. The column of Generations shows the number of elapsed generations estimated from the number of MA rounds and the recorded colony size of each round, while the mean and standard deviation of three replicates are presented. A list of all detected mutations and an extended table showing the number of intergenic BPS and short Indels are shown in S1 and S2 Tables, respectively.

| Strain | Abbr. | MA rounds | Generations | No. of Syn BPSs | No. of NSyn BPSs |
|---|---|---|---|---|---|
| MDS42 | WT | 66 | 1685.7 ± 7.0 | 0.67 ± 0.58 | 0.67 ± 0.58 |
| MDS42Δ*mutH* | H | 23 | 613.4 ± 1.7 | 4.33 ± 3.21 | 11.67 ± 1.53 |
| MDS42Δ*mutL* | L | 66 | 1675.5 ± 0.8 | 8.33 ± 2.31 | 15.00 ± 1.00 |
| MDS42Δ*mutS* | S | 66 | 1691.3 ± 9.3 | 11.00 ± 2.00 | 24.33 ± 1.15 |
| MDS42Δ*dnaQ* | Q | 66 | 1651.0 ± 5.9 | 8.67 ± 1.15 | 23.33 ± 2.89 |
| MDS42Δ*mutT* | T | 66 | 1679.7 ± 13.6 | 27.33 ± 2.31 | 157.67 ± 20.21 |
| MDS42Δ*mutL*Δ*mutH* | LH | 66 | 1651.4 ± 13.2 | 10.67 ± 1.53 | 15.00 ± 2.00 |
| MDS42Δ*mutS*Δ*mutH* | SH | 66 | 1691.1 ± 9.4 | 13.33 ± 5.13 | 21.00 ± 6.56 |
| MDS42Δ*mutL*Δ*dnaQ* | LQ | 69,61,61 | 1461.2 ± 96.0 | 545.67 ± 60.72 | 939.67 ± 81.5 |
| MDS42Δ*mutH*Δ*mutT* | HT | 23 | 610.8 ± 0.6 | 15.67 ± 5.03 | 76.33 ± 6.03 |
| MDS42Δ*mutL*Δ*mutT* | LT | 23 | 614.7 ± 2.5 | 16.00 ± 2.65 | 84.67 ± 14.57 |
| MDS42Δ*mutS*Δ*mutT* | ST | 23 | 605.0 ± 4.0 | 13.33 ± 4.51 | 54.33 ± 10.97 |
| MDS42Δ*dnaQ*Δ*mutT* | QT | 23 | 611.7 ± 2.6 | 13.33 ± 2.08 | 63.33 ± 7.09 |

In this analysis, the mutation rate was calculated based on the number of mutations observed at the endpoint of the MA experiments. This calculation assumes that the number of mutations accumulates linearly during the MA experiments. While the dynamics of mutation accumulation could potentially be more complex and variable, we adopt this simple assumption as it represents the least biased approach. Supporting this assumption, a previous study [26] demonstrated that during MA experiments spanning 6,000 generations, the mutation rates for the first 3,000 generations and the entire 6,000 generations were not significantly different. This observation is consistent with our assumption that the number of mutations increases linearly during our MA experiments.

As shown in Fig 1, our results showed that the mutator strains exhibited mutation rates approximately 6–400 times higher than that of the WT. The base-pair substitution patterns varied depending on the knockout gene(s), as shown in the pie charts in Fig 1, consistent with previous studies [24,27,30–33]. Specifically, disruption of the mismatch repair mechanism (Δ*mutS, ΔmutH, ΔmutL*) increased A:T to G:C and G:C to A:T substitutions and indels, whereas knockout of the *mutT* gene increased only A:T to C:G. Additionally, for each mutator strain, we calculated the dN/dS ratio, which represents the ratio of nonsynonymous to synonymous substitution rates. We found that the dN/dS ratios did not differ from 1 for all strains except WT, which had scarcely accumulated mutations (S2A Fig in S1 File). A dN/dS ratio close to 1 suggests that selection had little effect on our MA experiments.

We then quantified the growth rates of the mutator strains in M9 minimum medium without antibiotics. As depicted in S2B Fig in S1 File, the growth rate decreases with an increasing mutation rate, a pattern also observed in other studies [34]. This observed decrease in growth rate is thought to be caused, at least in part, by an increase in mutation rate. However, the potential contributions of other mechanisms, such as replication inhibition and activation of proofreading mechanisms, are also suggested [35].

## Quantifying speed of adaptation under antibiotics

To elucidate the relationship between mutation rate and evolutionary dynamics, we conducted experimental evolution of 12 mutator strains and the wild-type strain using five antibiotics

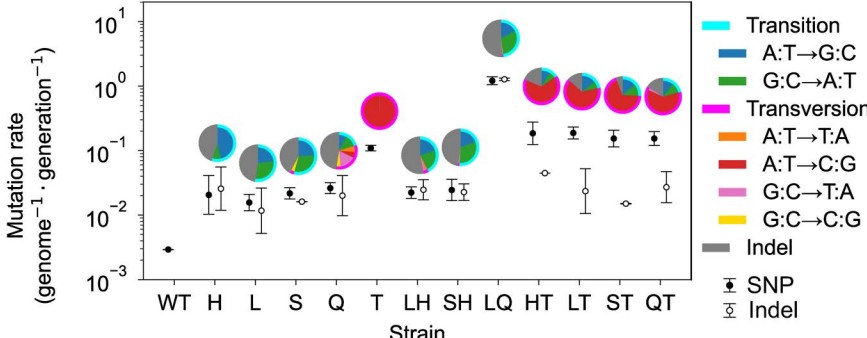

**Fig 1. Mutation rates of mutator *E. coli* strains.** Each dot with error bars represents the mean and standard deviation of the mutation rates observed in replicate MA lineages. The BPS rate (closed circle) was estimated using only synonymous mutations to exclude the effects of purifying selection. The indels rate (open circle) was also calculated only from the number of intergenic indels and normalized by dividing by the fraction of the length of the intergenic region in the genome (11%). The pie charts above the dots show the distribution of substitution patterns identified in the mutator strains, excluding the wild-type strain, which had as few as two mutations.

with distinct mechanisms of action, namely chloramphenicol (CP), trimethoprim (TP), amikacin (AMK), cefixime (CFIX), and ciprofloxacin (CPFX). We maintained four replica lines for each strain-antibiotic combination, resulting in a total of 260 individually evolving lines (13 strains × 5 antibiotics × 4 replicas). The cells were cultured in 200 µL of M9 medium supplemented with 20 amino acids (M9+AA medium) in a 96-well microtiter plate, to which each antibiotic was added in a two-fold dilution series (Fig 2A). Cultivation began with a fixed initial cell concentration ($OD_{620}$ value of $3×10^{-4}$, corresponding to approximately $2×10^5$ cells). After 24 h of incubation, cells were collected from the wells with the highest drug concentration among the wells with OD values above a certain threshold ($OD_{620}=0.03$). The collected cells were then transferred to fresh medium containing the antibiotic dilution series, with the initial $OD_{620}$ value of $3×10^{-4}$. We repeated this serial transfer procedure for nine days, corresponding to approximately 70 generations of cells. The number of generations during experimental evolution differed depending on the size of the overshoot from the threshold OD at each end of the incubation. S5 Table shows the precise number of generations of evolutionary lines calculated using OD values.

In this study, we defined the minimum inhibitory concentration (MIC) as the drug concentration in the well from which the cells were transferred. Fig 2B shows typical examples of the time series of MIC under TP selection, whereas all MIC time series of 260 evolutionary lines are presented in S3 Fig in S1 File. Using these time series MIC data, we evaluated the speed of drug resistance evolution under antibiotic treatment by determining the change in $log_2$-transformed MIC per day, which was obtained by comparing the MIC values of the first and last days of the 9-day evolutionary experiment. We assumed that MIC directly corresponds to fitness, disregarding the non-monotonic relationship between MIC and fitness as well as the dependency of the MIC-fitness relationship on drug concentration. We assumed that MIC directly corresponds to fitness, disregarding the non-monotonic relationship between MIC and fitness as well as the dependency of the MIC-fitness relationship on drug concentration. To assess the reproducibility of our observations, we conducted a replicate experiment over a shorter period of five days. Except for AMK, which exhibited relatively small absolute values of MIC doubling rates, the results demonstrated clear correlations between the adaptation speeds in the 9-day and 5-day experiments (S4 Fig in S1 File). This emphasizes the reproducibility of our adaptation speed estimates.

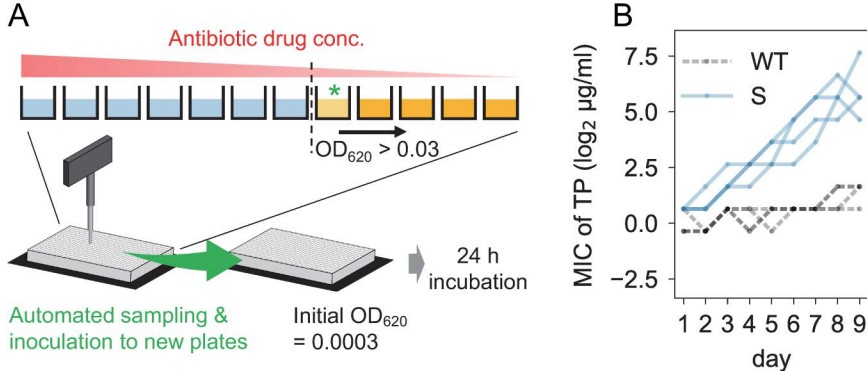

**Fig 2. Experimental evolution under antibiotic selection pressure.** (A) Schematic representation of the experimental procedure. A 96-well microtiter plate was prepared with two-fold serial dilutions of antibiotics. After 24 hours of cultivation, cells from the well with the highest drug concentration, in which the cell concentration (OD620) exceeded 0.03, were transferred to fresh medium with a drug concentration gradient. In this study, we designated the drug concentration of the selected well (green asterisk) as the minimal inhibitory concentration (MIC). The serial transfer cultures were iterated for 9 days. (B) Example of MIC changes during experimental evolution. The vertical axis shows the MIC for trimethoprim (TP). The black and blue lines represent the data for the WT and *ΔmutS* (S) strains, respectively. For each strain, data from four replicate series are overlaid. Data for all combinations of strains and drugs are presented in S3 Fig.

Fig 3 illustrates the relationship between the mutation rate and the rate of adaptation for each antibiotic. In this analysis, the mutation rate was calculated based on the frequency of BPSs. As observed, the rate of evolutionary adaptation generally increased with mutation rate, except for the LQ strain, which had the highest mutation rate and often exhibited significantly slower adaptation speeds. A similar pattern is also observed when the mutation rate is calculated using the combined frequency of BPSs and indels (**S5 Fig** in **S1 File**).

The dependency on mutation rate varied with the antibiotics used for selection. For CP and TP selections, significant decreases in MIC doubling rates were observed in the LQ strain compared to the other five strains with mutation rates close to 0.1 (p<0.01; Mann-Whitney U-test). Conversely, no significant decreases were observed for the other three drugs. CP and TP are known as bacteriostatic drugs, while AMK, CFIX, and CPFX are classified as bactericidal drugs. Thus, our results suggest that the effect of mutation rate on antibiotic resistance evolution depends on the mechanisms of action of the drugs used for selection.

**Analysis of mutations fixed in evolved *E. coli* populations.** To investigate whether the profiles of mutated genes vary depending on mutation spectra, we sequenced the genomes of evolved resistant populations. Specifically, we analyzed 24 populations after nine days of antibiotic selection, encompassing three mutator strains (L, Q, and T), two antibiotics (CP and CFIX), and four independently evolved populations for each condition. The fixed mutations identified in these populations are summarized in S7 Table.

Our analysis revealed that the mutation spectra of these fixed mutations were consistent with those observed in mutation accumulation experiments (S6 Fig in S1 File). For example, A:T to C:G mutations were significantly enriched in the *mutT* deletion strain, aligning with its known mutational signature.

Furthermore, this analysis demonstrated that mutations commonly associated with antibiotic resistance evolution are also prevalent in mutator populations. Under CFIX selection, mutations in *ampC*, which encodes β-lactamase, and *ompR,* which is involved in the regulation of porin protein, were detected in multiple evolved populations. These mutations are well-documented for their role in conferring resistance to β-lactams, including CFIX [36,37].

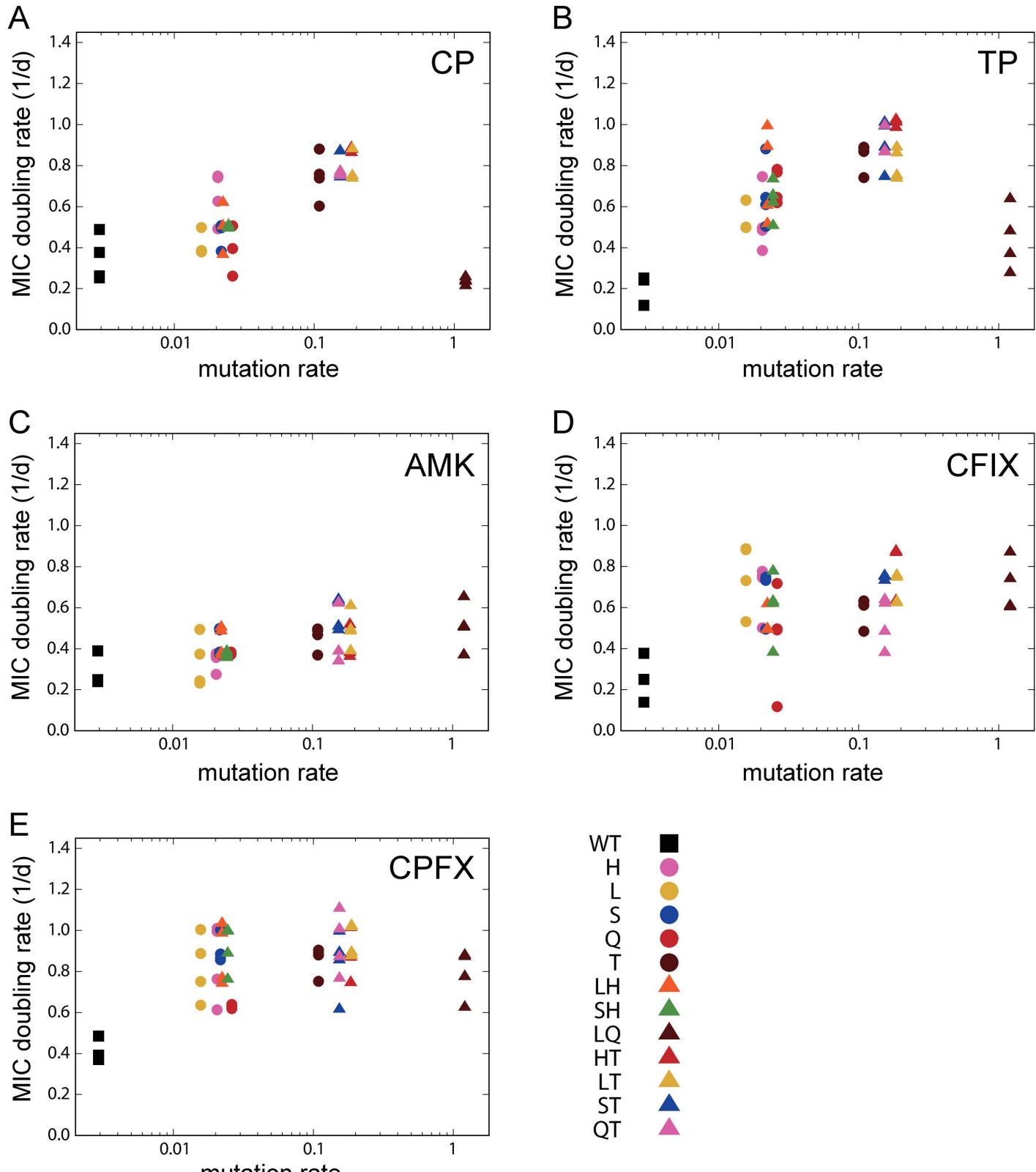

**Fig 3. The relationship between mutation rate and MIC doubling rate.** The dots represent experimental observations of 13 strains and four replicate serial transfer cultures. To prevent overlap of data points, small Gaussian noise (mean = 0, standard deviation = 0.05) was added to the y-coordinates.

Similarly, in populations evolved under CP selection, mutations were frequently identified in *acrB*, which encodes a multidrug efflux pump, as well as in *acrR*, *marR*, and *soxR*, which regulate its expression. These mutations are also known to contribute to resistance against a broad spectrum of antimicrobial agents [38].

**Population dynamics model explains the relationship between mutation rate and speed of adaptation.** To elucidate the mutation rate dependency of adaptation speed, we employed a simple population dynamics model of multistep resistance evolution that incorporates pharmacodynamic modeling [39,40]. In this model, *E. coli* populations can grow and increase their cell number up to a certain carrying capacity, while accumulating mutations during their replication at a given mutation rate. Specifically, we considered the sequential accumulation of mutations that confer additive beneficial effects on growth in the presence of antibiotics (Fig 4A). Population dynamics are described by the following deterministic differential equations:

$$\frac{dM_i}{dt} = \mu p_b M_{i-1}\left(1 - \frac{M}{K}\right) + \mu(1 - p_b)M_i\left(1 - \frac{M}{K}\right) - \gamma_i M_i \tag{1}$$

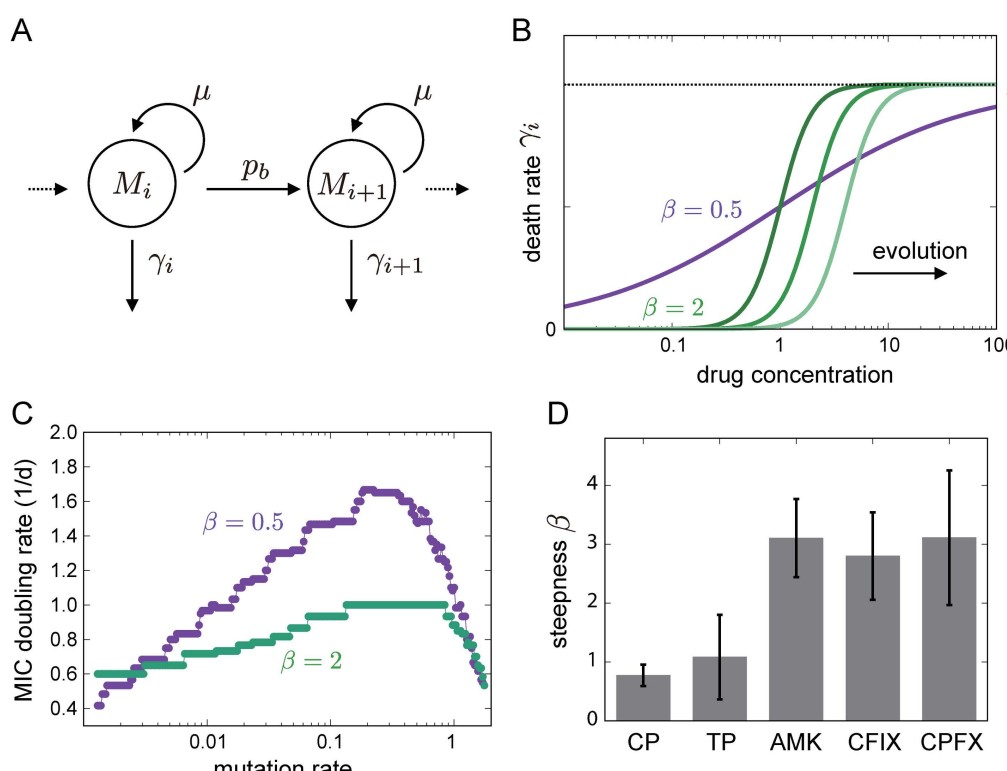

**Fig 4. Parameter estimation by a population dynamics model.** (A) Schematic representation of the model with multi-step resistance evolution. The mutated subpopulations ($M_i$) with $i$ mutations proliferate with the growth rate $\mu$, mutate with the rate $p_b$, and die at the death rate $\gamma_i$. The death rate depends on the benefit of mutation $b_i$ conferred per mutation, as represented in Eq. (3). (B) The effect of antibiotics on the death rate. For the cases of $\beta = 0.5$ and $\beta = 2$, the death rates are plotted as a function of drug concentration, where $b_0$ is set to 1. The death rate curve shifts towards the right due to selection, as depicted by the right green lines in the case of $\beta = 2$. (C) The simulated relationship between mutation rate and MIC doubling rate. The population dynamics model with mutation accumulation was utilized to simulate the increase in MIC observed in serial transfer experiments for the cases of $\beta = 0.5$ and $\beta = 2$. The parameters were set to $K = 0.2$, $b_0 = 0.5$, and $\in = 0.5$, respectively. (D) The estimated beta value for each drug. The mean value of $\beta$ estimated through 100 bootstrap resampling is represented by the bars, while the error bars represent the standard error.

with $M = \sum_i M_i$. Here, $M_i$ represents the population size of *E. coli* cells that have accumulated $i$ beneficial mutations, $\mu$ is the growth rate, $\gamma_i$ is the death rate with $i$ mutations, and $K$ is the carrying capacity, respectively. The variables and parameters used in this model are summarized in Table 2. The probability of acquiring a beneficial mutation, $p_b$, is assumed to be proportional to the mutation rate, $p$, such that $p_b = \alpha p$, where $\alpha$ is the ratio of beneficial mutations to total mutations. The first term on the right side of Eq. (1) represents the influx of population caused by a single beneficial mutation. For simplicity, we do not allow reverse mutations that decrease the number of beneficial mutations. Note that the expected number of mutations during the 9-day evolution experiment with the LQ strain, which had the highest mutation rate among the mutator strains we used, was ~70. Therefore, the chance of reverse mutation, that is, positional overlapping of these ~70 mutations on the genome size of 4Mbp, was very low.

By measuring the growth rate of the constructed mutant strains, we observed that the growth rate $\mu$ decreased as the mutation rate $p$ increased, as shown in S2B Fig in S1 File owing to the deleterious effects caused by the gene deletion. We fit the data in S2B Fig in S1 File with the following relationship:

$$\mu = \mu_0 \exp(-\delta p) \tag{2}$$

and obtained $\mu_0 = 0.90$ per hour and $\delta = 0.54$, which represents the maximum growth rate and the coefficient for deleterious effect of mutations on cellular growth. Note that the observed reduction in growth rate may not be fully explained by high mutation rates alone. Although the mechanism for the growth defects cannot be quantified by mutation fixation rates, for simplification, we have assumed that the contributions of these non-observable factors are proportional to the observed mutation rate. Under this assumption, Eq. (2) describes the relationship between population growth rate and mutation rate, incorporating both the

**Table 2. Summary of variables and parameters used in the population dynamics model.**

| Symbol | Description |
| --- | --- |
| $M_i$ | population size of cells having $i$ beneficial mutations |
| $M$ | total population size, $M = \sum_i M_i$ |
| $\mu$ | growth rate of cells, $\mu = \mu_0 \exp(-\delta p)$ |
| $\gamma_i$ | death rate of cells with $i$ mutations |
| $K$ | carrying capacity of environment |
| $p$ | mutation rate |
| $\alpha$ | ratio of beneficial mutations to total mutations |
| $p_b$ | beneficial mutation rate, $p_b = \alpha p$ |
| $\mu_0$ | maximum growth rate |
| $\delta$ | coefficient for deleterious effect of mutations on cellular growth |
| $x$ | $\log_2$-transformed concentration of antibiotics |
| $\beta$ | sensitivity of the death rate to the addition of antibiotics |
| $b_i$ | resistance of cells having $i$ beneficial mutations |
| $b_0$ | resistance of non-mutated cells |
| $\varepsilon$ | benefit of a single cell on resistance |

effects of high mutation rates and non-observable factors. We then used these fitted parameters for the numerical simulations. Furthermore, for simplicity, we designed our simulation such that the evolutionary variable of a strain was only drug resistance. Therefore, the growth rates of the strains were maintained throughout the simulations.

In our model, the addition of antibiotics resulted in an increase in the cell death rate. However, the acquisition of beneficial mutations can enable cells to overcome the effects of drugs. We assumed that the combined effect of antibiotics and beneficial mutations can be represented by the following death rate:

$$\gamma_i = \mu \frac{\left(x/b_i\right)^\beta}{\left(x/b_i\right)^\beta + 1} \tag{3}$$

where $x$ is the $\log_2$-transformed concentration of antibiotics and the parameter $\beta$ represents the sensitivity of the death rate to the addition of antibiotics (see Fig 4B). At sufficiently high drug concentrations, the death rate $\gamma_i$ converges to the growth rate $\mu$, and the cells are unable to proliferate further. We do not consider the case where $\mu < \gamma_i$, as a subpopulation with $\mu < \gamma_i$ is rapidly diluted in our serial transfer culture and becomes negligible. $b_i$ represents the benefit resulting from the accumulation of $i$ mutations, which corresponds to the $\log_2$-transformed drug concentration that reduces the growth rate of the subpopulation by half (close to the MIC in our experimental setting). In bacterial experimental evolution under antibiotics, the MIC often increases exponentially (see S3 Fig in S1 File and [41,42]). To model the exponential increase of MIC by mutations, we simply assume the following additive effect of mutations on the benefit: $b_i = b_0 + i\epsilon$ , where $b_0$ and $\epsilon$ are constant parameters representing the resistance of non-mutated cells and the benefit of a single mutation, respectively. Note that the unit of benefit $b_i$ is the log-transformed concentration; thus, the above additive effect of the mutations represents an exponential increase in the MIC. We simplified our model by neglecting the individuality of mutations and assuming that each mutation has the same beneficial effect and cost. Additionally, we neglected any positive or negative epistasis between the mutations.

We utilized a population dynamics model to simulate the adaptive evolution of antibiotic resistance in our serial transfer setup (Fig 2). In the initial condition, non-mutated cells were inoculated into a series of 2-fold dilutions of antibiotics with a constant initial density denoted as $M_0 = 3\times10^{-4}$ and $M_i = 0$ for $i \neq 0$, in units of $OD_{620}$ which corresponds to approximately $2\times10^5$ cells in our experimental settings. After 24 h of cultivation, cells were sampled from an environment with the highest drug concentration, in which the total cell density exceeded a certain threshold ( $M = \sum_i M_i > 0.03$), diluted, and subsequently transferred to a fresh environment for the next round of cultivation. This simulation procedure corresponds to our experimental design (Fig 2A) and was iterated nine times, resulting in an increase in minimum inhibitory concentration (MIC) similar to that observed in Fig 2B.

Fig 4C presents typical simulation results for the MIC doubling rate as a function of the mutation rate. As shown in the figure, the curve is significantly influenced by $\beta$, which represents sensitivity to antibiotics. When $\beta$ is equal to 0.5, the MIC increases significantly with an increase in the mutation rate before experiencing a drastic decrease when the mutation rates become extremely high. Conversely, when the sensitivity was high, as indicated by the steeper kill curve in Fig 4B, the change in the MIC with increasing mutation rates was smaller. This difference can be explained by the variation in effective population sizes between the two conditions. In the case of death rate with low steepness, cells with fewer beneficial mutations have a greater chance of growth, and these cells can also acquire further beneficial mutations. Therefore, the speed of adaptation can increase substantially with an increase in the mutation

rate. In contrast, when the sensitivity curve is steep, only a small fraction of cells with beneficial mutations can grow, and MIC improvement is primarily governed by the time required for the mutated cells to take over the population. Hence, in this case, the effect of the increased mutation rate on the speed of MIC improvement was relatively small.

We used this population dynamics model to fit the experimental data and to elucidate the evolutionary dynamics presented in Fig 5. The experimentally estimated values of $K$, $\mu_0$, and $\delta$ were used as fixed parameters, while $\epsilon$, $b_0$ and $\beta$ were used as fitting parameters to minimize the residual sum of squares between experimental and simulated adaptation speeds for each antibiotic. Because of the difficulty in experimentally estimating parameter $\alpha$, which represents the ratio of beneficial mutations, we assigned an arbitrary value and confirmed that the simulated results remained qualitatively unchanged within a certain range. The green solid lines in Fig 5 represent the average of the simulated results under the fitted parameters, with the light green range indicating the standard error estimated by bootstrap random resampling of the experimental data.

Our analysis revealed an interesting finding that the estimated sensitivity $\beta$ significantly differed among antibiotics (Fig 4D), with relatively low values for CP and TP and high values for the other three antibiotics. This result is consistent with the classification of bacteriostatic and bactericidal antibiotics, where the former is expected to have shallow killing curves. However, interpreting the estimated beneficial effect of each mutation represented by $\epsilon$ (S7 Fig in S1 File) remains difficult and may be related to the physical characteristics of the drugs. For instance, drugs with high estimated $\epsilon$ values typically have smaller molecular weights (e.g., the molecular weights of CP, TP, and CPFX are approximately 300), whereas those with lower values tend to have larger molecular weights (e.g., the molecular weight of AMK is 586 and that of CFIX is 453). Regarding the observed correlation between $\epsilon$ values and the molecular weight of antibiotics, we propose the following hypothesis: One of the main mechanisms for acquiring antibiotic resistance in this experimental setting involves the activation of efflux pumps. The activity of these pumps may be influenced by the molecular weight of the drugs, implying that mutation effects might similarly depend on molecular weight. Currently, there is no experimental data to substantiate this relationship; however, we believe these parameter estimates offer a novel perspective for further analysis.

In our model simulations, with $\beta = 0.5$, as observed with drugs such as CP and TP, doubling the drug concentration was predicted to have a minimal impact on the population growth rate, suggesting that most cells retain their ability to proliferate (S8 Fig in S1 File). Conversely, at $\beta = 4$, which is a characteristic of bactericidal drugs, a doubled drug concentration significantly decreased the growth rate, indicative of a decrease in population size. Such variations in population dynamics in response to changes in drug concentration may influence the observed mutation-rate dependency on the speed of adaptation.

## Discussion

In this study, we conducted quantitative analysis to investigate how changes in mutation rates affect the evolution of antibiotic resistance in *E. coli*. To achieve this goal, we generated a library of *E. coli* mutator strains using single and double deletions of genes related to DNA replication and error correction. We quantified the mutation rates of these mutators using MA experiments, which resulted in a range of mutation rates caused by gene deletions. We then performed evolution experiments starting from these mutators by serial transfer culture under five antibiotics and quantified the rate of MIC increase over time. The results demonstrated that, for nearly all mutator strains, the rate of adaptation positively correlated with mutation rate. However, an exception was observed in the LQ strain, which exhibited the highest

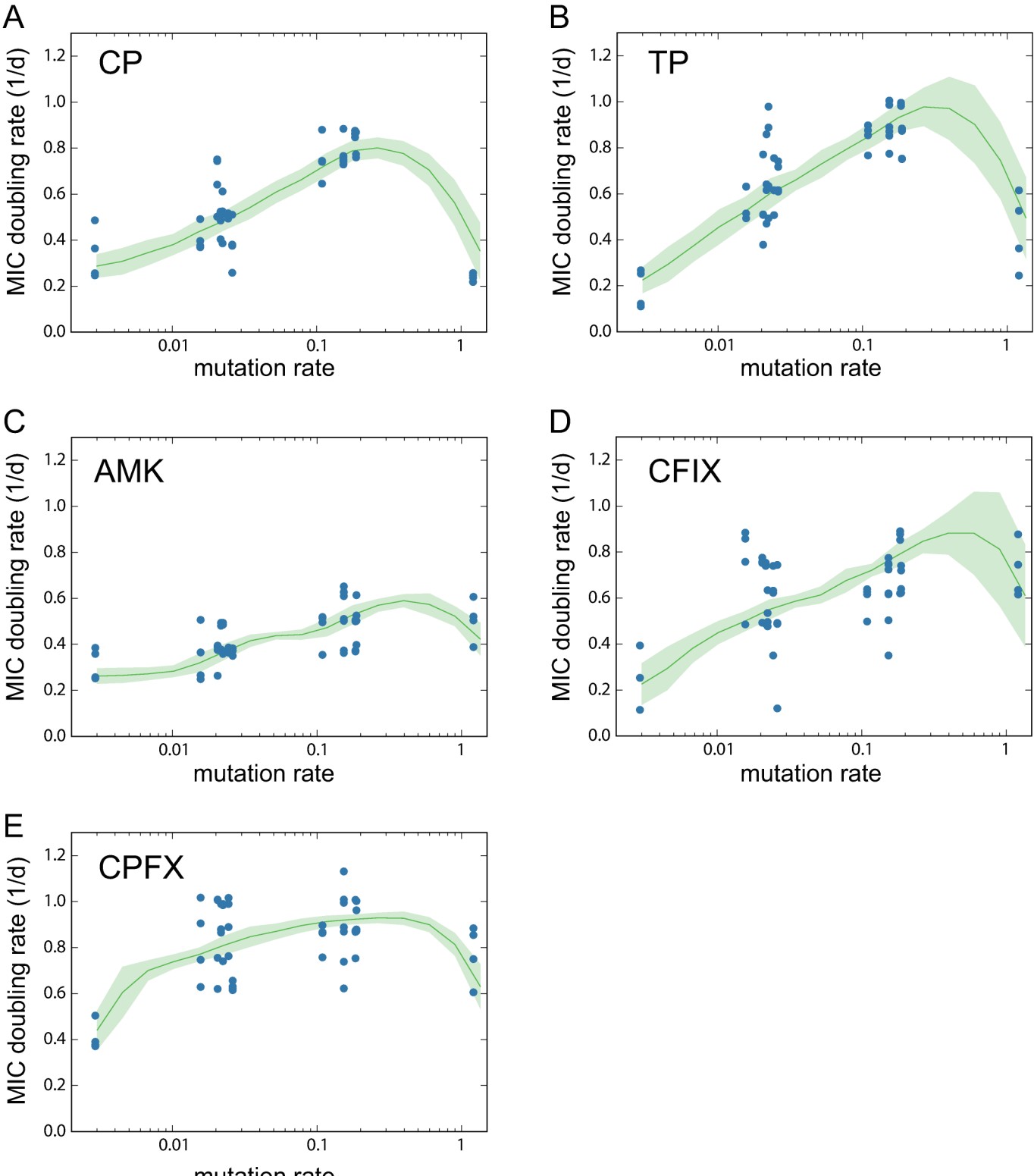

**Fig 5. The fitting results of the population dynamics model to experimental observations.** The blue dots represent experimental observations shown in Fig 3 and the green solid line shows the mean of simulated MIC changes generated with the parameters estimated from 100 bootstrap resampling, while the light green region represents the estimated standard error.

mutation rate but frequently showed a significant reduction in the adaptation rate (Fig 3). This finding highlights that the relationship between mutation rate and evolutionary adaptation rate is potentially non-monotonic under certain conditions.

The observed reduction in the adaptation rate of the LQ strain is associated with its growth defect (S2B Fig in S1 File). However, the underlying mechanism responsible for this growth defect remains unclear. One possible explanation is that the simultaneous deletion of the *mutL* and *dnaQ* genes exerts a synthetic effect, thereby impairing growth. Alternatively, this study posits a model in which the accumulation of deleterious mutations and other detrimental effects, proportional to the mutation rate, ultimately leads to a reduction in growth rate. Both hypotheses are equally plausible and provide potential explanations for the observed decrease in the evolutionary rate due to growth impairment. However, no experimental data are currently available to distinguish between these two possibilities.

Nevertheless, our population dynamics model, which assumes the accumulation of deleterious mutations, provides a general framework for describing the relationship between growth inhibition and evolutionary rate. This approach may offer a suitable method for characterizing the interplay between mutation rate and evolutionary adaptation rate. Furthermore, previous studies investigating the relationship between mutation rate and growth rate across various mutator strains and culture conditions have consistently demonstrated that an elevated mutation rate frequently results in a decline in growth rate [34]. These observations suggest that a spectrum of deleterious mutations may collectively contribute to the observed growth defects.

Our results indicate that the relationship between the speed of adaptation and increasing mutation rates was dependent on the selective agent. When using bacteriostatic drugs, a modest mutation rate led to a significant increase in the speed of adaptation, whereas bactericidal drugs exhibited less dependency of adaptation speed on the mutation rate. This difference provides novel insights into the conditions for the emergence of resistant strains. In cases where a two-fold change in antibiotic concentration used in the evolution experiment has a relatively small effect on lethality, that is, if it has a lower $\beta$ coefficient in our killing curve model (Eq. (3)), a greater number of cells have survived after the drug increase and obtained the opportunity to acquire beneficial mutations, resulting in stronger selection favoring high mutation rates under the drug treatment. Conversely, if sensitivity to a drug shows a steep response, only a small fraction of cells survive and have a chance to acquire beneficial mutations. In this case, with a smaller effective population size, the benefit of increasing the mutation rate for evolution of resistance is limited. In both cases, with extremely high mutation rates, such that each individual in a population obtains one or more mutations, selection cannot remove deleterious mutations regardless of the population size. This can be a possible scenario that the LQ strain showed little or no advantage in MIC improvement. While the relationship between the mutation rate and the speed of adaptive evolution has been previously explored [5,13], these analyses were conducted under a single selection environment. In contrast, the current study employed various drugs with distinct action mechanisms, implying that the effective population size under the influence of drug selection may impact the association between mutation rate and adaptation speed. Notably, our results showed that the speed of adaptation to amikacin addition did not increase significantly with higher mutation rates. This result may shed light on the mechanism responsible for the infrequent emergence of amikacin-resistant strains [43].

There may be alternative explanations for the observed differences in adaptive evolution under bacteriostatic and bactericidal drugs. For example, the size or nature of the mutational targets leading to resistance acquisition may differ among these drugs, as discussed in previous studies [44,45]. Although our previous research demonstrated that the number of fixed mutations did not significantly differ among evolved *E. coli* strains under these drugs [42], a

more detailed analysis of the fixed mutations from the evolved strains obtained in this study could provide valuable insights into the differences in the evolution of resistance to bacteriostatic and bactericidal agents.

Recently, Lukačišinová et al. conducted drug-resistance evolution experiments in *E. coli* using an automated platform to quantify the evolvability of 98 strains with gene deletions [46]. Their study demonstrated epistasis between resistance mutations and the genetic background by identifying gene deletion strains that affect the evolution of drug resistance. The gene deletion library they used included *mutL* and *mutT*, which showed a moderately high mutation rate in our MA experiment. Additionally, they used Tetracycline and Chloramphenicol, which are bacteriostatic drugs. Their experiment also showed that deletion of *mutL* and *mutT* generally accelerated the evolution of resistance to these drugs, supporting our conclusion. Note that they selected gene deletions by ensuring that they did not bring significant growth reduction to the cells in drug-free media. This gene selection contrasts with our research which focused on the impact of higher mutation rates alongside growth reduction on evolvability. Thus, their study elaborates on the effect of gene deletions mainly involved in efflux pumps on drug resistance and resistance evolution, as well as the types of mutations that occur in the evolved strain. On the other hand, our numerical approach was able to quantify various factors, such as mutation rate, growth rate, and features of the selective environment, and to predict the effect of changes in the mutation rate on the evolution of resistance for any known selective agent.

As shown in Fig 1, the results of MA experiment indicate that A:T→C:G mutations become predominant due to the deletion of the *mutT* gene. Such changes in the mutation spectrum could potentially affect the rate of evolution. Indeed, the evolved strains with a relatively high evolutionary rate, approximately a mutation rate of 0.1, are those lacking the *mutT* gene, as demonstrated in Fig 3. The frequent occurrence of A:T→C:G mutations in these strains may contribute to their high evolutionary speeds. To explore this possibility, we analyzed fixed mutations in antibiotic-resistant *E. coli* strains obtained in a previous study of experimental evolution [42]. We hypothesized that if A:T→C:G mutations significantly contribute to resistance acquisition, an enrichment of these mutations in antibiotic-resistant strains would also be expected in the evolutionary dynamics of non-mutator strains. However, as shown in S8 Table, A:T→C:G mutations are not consistently enriched; instead, the distribution of mutations aligns with the base-pair substitution profile of wild-type *E. coli* [26]. This suggests that A:T→C:G mutations in *mutT* deletion strains do not necessarily increase the likelihood of resistance enhancement or accelerate the evolution of drug resistance.

The analysis of the mutational spectrum also revealed that the ratio of indels to BPSs varied depending on the knockout gene(s). As shown in Fig 1, when the LQ strain is excluded as an outlier, mutator strains with a high fixation rate of BPSs do not necessarily exhibit a correspondingly high frequency of indels. Given that indels often have a more pronounced impact on phenotypic changes compared to BPSs and are more likely to result in lethal mutations, substantial variation in the ratio of indels to BPSs across different mutation rates may influence the evolutionary dynamics of distinct mutator strains. Given the limited amount of data and the challenges associated with conducting statistical analyses, it is difficult to draw definitive conclusions from this study. Future research with sufficient data should be conducted to further explore the relationship between the mutational spectrum in mutators and the evolution of drug resistance.

The *mutL* deletion strains exhibited lower growth rates compared to mutator strains with similar mutation rates (S2B Fig in S1 File), suggesting that the growth reduction in these strains cannot be solely attributed to mutation rate. The underlying mechanism of the slow growth observed in *mutL* deletion strains remains unclear. The mechanism underlying the

slow growth observed in *mutL* deletion strains remains unclear. One possible explanation involves an imbalance between the functions of the MutL and MutS proteins. As reported [47], MutL and MutS coordinately contribute to mismatch repair; MutS binds specifically to mismatched DNA, while MutL helps to release the bound MutS. This suggests that maintaining a proper balance between MutS and MutL is critical for efficient mismatch repair. The deletion of the *mutL* gene may disrupt this balance, leading to an excess of unbound MutS, which could contribute to the observed growth defect.

In the experimental evolution under antibiotics, there exists a possibility that selection may favor reversing growth defects in ancestral strains rather than enhancing antibiotic resistance [46]. To investigate this in our experimental evolution with mutator strains, we measured the growth rates of three strains (WT, S, and LQ) both before and after experimental evolution. As presented in S9 Fig in S1 File, the growth rates remained largely unchanged. This finding strongly suggests that the observed increases in MIC during our experiments were not attributable to recovery in growth rates but were likely driven by adaptations related to antibiotic resistance.

The experiments and simulations presented in this study encompass numerous limitations and unvalidated hypotheses. For instance, in our population dynamics model, we assumed $b_i = b_0 + i\varepsilon$, which means that the log-transformed MIC linearly increases with the number of beneficial mutations. This assumption is grounded in observations from our nine-day experimental evolution shown in S3 Fig in S1 File, where the log-transformed MIC increased approximately linearly over time across many culture lines. However, for some lineages, the log-transformed MIC shows a saturation curve, indicating that mutations fixed later have a diminishing effect. This trend has been observed in several experimental evolution studies, including Lenski's long-term experimental evolution [48]. To account for this, we have modified our model to include a saturation curve for the increase in fitness due to mutations, as $b_i = b_0 + i\varepsilon S / (i + S)$. Here, $S$ is a parameter that influences the shape of the saturation curve. When $S$ is small, the increase in log-transformed MIC saturates with only a few mutations, whereas a larger $S$ results in an almost linear increase, akin to the original model. Using this modified model, we estimated other parameters using a genetic algorithm. S10 Fig in S1 File illustrates how the average residuals between the model predictions and observed data points change when parameter $S$ is fixed. As depicted, the residuals decrease as $S$ increases. This result supports the appropriateness of using the assumption in the original model, which corresponds to the case of large $S$.

In addition to the examples mentioned above, there are various other assumptions that require verification. For instance, we assume that the mutation rate remains constant throughout the experimental evolution, although this assumption lacks empirical verification. Previous studies have demonstrated that mutations that alter the mutation rate can be fixed during mutation accumulation experiments [49]. For other examples, the functional forms of equations (1) and (2) assumed in the simulation, the constancy of $\epsilon$, and the consideration of only the death rate when using a common growth rate should be subjected to future experimental validation.

The evolution of mutation rates and the effects of new mutations on reproductive success have been well studied [2,50] while pathogen/drug resistance has been used merely as a marker trait to estimate mutation rates using the fluctuation test [51,52]. Our study builds on previous observations of the relationship between the mutation rate and the speed of adaptation by quantifying specific features of a selective environment that controls the shape of this relationship. Further experimental and theoretical investigations would help to better understand not only the evolution of drug-resistant microbes but also other evolutionary challenges, such as the emergence of drug-resistant cancer cells [53].

## Methods

### Bacterial strains and media

The insertion sequence (IS)-free *E. coli* K-12 substrain MDS42 [54], purchased from Scarab Genomics (Madison, Wisconsin, USA), was used as the wild-type (WT) strain in this study. Using this WT strain, we constructed 12 knockout strains of genes related to the suppression of mutations (*mutS, mutH, mutL, mutT, dnaQ*), as listed in Table 1. Single- and double-gene knockouts were performed using the λ-Red homologous recombination method [55]. In this study, we utilized an IS-free strain to ensure reliable resequencing analysis, as determining the precise positions of IS element insertions can be challenging when using short-read sequencing technologies. *E. coli* cells were cultured in modified M9 minimal medium supplemented with 20 amino acids (M9+AA medium) which includes 17.1 g/L $Na_2HPO_4 \cdot 12H_2O$, 3.0 g/L $KH_2PO_4$, 0.5 g/L NaCl, 2.0 g/L $NH_4Cl$, 5.0 g/L glucose, 14.7 mg/L $CaCl_2 \cdot 2H_2O$, 123.0 mg/L $MgSO_4 \cdot 7H_2O$, 2.8 mg/L $FeSO_4 \cdot 7H_2O$, and 10.0 mg/L thiamine hydrochloride (pH 7.0), and 20 canonical amino acids (0.02 mM Tyr and 0.05 mM of the remaining 19 amino acids: Gly, Ala, Asn, Gln, Leu, Ile, Met, Ser, Pro, Thr, Val, Phe, Trp, Asp, Glu, Arg, His, Lys and Cys) [34]. In this study, to enhance the reproducibility of cultures and facilitate future analysis of medium components during resistance evolution, we opted for a synthetic medium with defined components rather than a rich medium like LB, which contains unspecified components. Moreover, to improve the robustness of *E. coli* against mutations [34], we employed a minimal medium supplemented with amino acids.

### Measurement of growth rate

To measure the growth rate, cells precultured to yield an initial $OD_{620}$ of 0.001 were inoculated into a 96-well microtiter plate containing 200 μL M9+AA medium. The cells were then cultured at 34 °C with shaking at 300 rpm, and the cell density was measured at $OD_{620}$ using a FilterMax F3 microplate reader (Molecular Devices) at 30-minute intervals. The specific growth rate was calculated as the slope of the exponential growth curve determined by data points with $OD_{620}$ values under the condition $0.005 < OD_{620} < 0.03$, according to the standard Malthusian growth model.

### Mutation accumulation (MA) experiment

*E. coli* cells were subcultured as single colonies on M9+AA agar medium and incubated at 34 °C. A colony was randomly selected and streaked onto fresh agar medium every two days for a total of 23 or 61-69 passages. The number of generations per passage was estimated based on colony diameter, which has been shown to be a reliable indicator [26–28]. Specifically, we used a regression curve to estimate colony-forming units from the diameter and then calculated the number of generations based on the assumption that all cells in a colony were derived from a single cell from the previous passage. The adopted relationship used in this study was (number of generations) = 24.51 + 2 $\log_2$ (colony diameter in mm), which is a reuse of that experimentally obtained in our previous study [28]. After each passage, we cultured the remaining cells of the selected colony in M9+AA liquid medium and stored them in 15% glycerol. We kept three MA lines for each ancestral strain, resulting in 39 MA lines. S3 Table shows the colony size for each round recorded during the MA experiment.

### Whole-genome resequencing

Glycerol-stocked cells were grown in M9+AA medium until they reached full growth, and were then harvested as a cell pellet by centrifugation. Genomic DNA was extracted from cells

using a DNeasy Blood & Tissue Kit (Qiagen). For library preparation prior to sequencing, a Nextera XT kit (Illumina) was used in a paired-end (2×300 bp) setting. The Illumina MiSeq platform was used to sequence the libraries using the MiSeq Reagent Kit v3, which provided 600 cycles (Illumina). The obtained short reads were mapped to the reference genome (GenBank accession: AP012306.1), and point mutations accumulated during the MA experiment were detected using Breseq [56] (Ver.0.30.1) with bowtie2 (Ver.2.1.0) and R (Ver.3.2.4). The known common mutations in our ancestral wild-type MDS42 cells (S4 Table) were omitted from the detected mutations in this study. Also, the mutations in the genome of each mutator ancestor cell in the MA experiment was omitted from the detected mutations in the post-MA cell lineages.

## Calculation of mutation rate and dN/dS

We calculated the base-pair substitution (BPS) rate as the mutation rate using the numbers and spectra of synonymous mutations accumulated during MA and codon-usage information on the ancestral genome, following a previous publication [29]. First, we classified the accumulated synonymous substitutions into each of the six possible types, and divided their numbers by the number of sites in the genome where each type of substitution is synonymous, resulting in the number of substitutions per site, excluding the selective effect. Then, we multiplied the number of synonymous, nonsynonymous, and non-coding BPS sites by the number of substitutions per site and summed the results for the six types of substitutions. Finally, we divided this by the number of MA generations to obtain the genome-wide base substitution rates. To obtain dN/dS, we first calculated dN by dividing the number of nonsynonymous substitutions by the number of nonsynonymous sites, which was normalized considering the spectrum of BPS in the same manner as calculating the BPS rate. Similarly, we calculated dS by dividing the number of synonymous substitutions by the number of synonymous sites, which was normalized considering the spectrum of BPS.

## Experimental evolution under antibiotics

The cells were cultured in 200 μL of M9+AA liquid medium in a 96-well microtiter plate. During the experiment, each culture line was exposed to 12 concentrations of antibiotics, corresponding to 11 wells with a two-fold dilution series, and one drug-free well. For CP, the well with the highest drug concentration was 600 ng/μl. Similarly, 200 ng/μL TP, 400 ng/μL AMK, 40 ng/μL CFIX, and 4 ng/μL CPFX. Cell concentration was quantified by measuring the optical density at 620 nm ($OD_{620}$). Cells were inoculated into these 12 wells at an initial $OD_{620}$ of $3×10^{-4}$. After 24 h of cultivation at 34 °C with shaking at 300 rpm, cells were sampled from the well with the highest drug concentration among the wells that showed an $OD_{620}$ value greater than 0.03. Cultivation at 34°C was chosen to reduce the growth rate compared to the standard condition of 37°C, thereby facilitating subculturing during the exponential growth phase in our experimental settings. The sampled cells were diluted to an initial $OD_{620}$ value of $3×10^{-4}$ and inoculated into fresh medium in 12 wells of a new plate, which was then incubated. Four independent culture lines were maintained in parallel for each combination of strain and antibiotic. A serial transfer experiment was conducted for nine days. After that we stored the evolved samples in 15% glycerol at -80°C. Some of the samples were then subjected to whole-genome resequencing analysis in the same manner as the samples from the MA experiment. The described experimental operations were performed using an automated culture system comprising a Biomek NXP laboratory automation workstation (Beckman Coulter) in a clean booth, STX44 automated shaker incubator (LiCONiC), LPX220 plate hotel (LiCONiC), and a FilterMax F3 microplate reader (Molecular Devices) [57].

## Parameter estimation

Custom C programs were used to implement the population dynamics model described in underlined equation (1) and optimize the model parameters. Specifically, a genetic algorithm was employed to estimate the values of the parameters $\epsilon$, $b_0$ and $\beta$. The fitness function for the genetic algorithm is defined as the Euclidean distance between the experimentally observed MIC doubling rate and the rate obtained from simulations of t population dynamics model. The population size for the genetic algorithm was set to 100, and the parameter sets with the top 5% highest fitness were selected. each generation, a random change of 0.5% was applied to each parameter. The mean and standard errors of the parameters were estimated using bootstrap resampling of the experimental data, wherein the same number of experimental data points was resampled with allowed duplication.

## Supporting information

**S1 File.   S1 Fig.** Mutator Construction Procedure. Each box represents either a mutator strain or the wild-type strain. Arrows depict the lineage from parent to offspring in the construction of mutators. The list of mutations on the right details those shared between each parent-child pair (refer to S1 Table for the complete list of mutations). The values indicated in each strain box represent the growth rates (1/h; mean ± SD). **S2 Fig.** (A) Neutrality in mutation accumulation. The dN/dS ratio was calculated for each mutator strain and the wild-type strain. Error bars show the standard deviations between the MA lineages. (B) Relationship between growth rate and mutation rate. The horizontal error bars show the standard deviation across MA lineages, whereas the vertical error bars represent the standard deviation among replicate experiments in growth rate measurement. The sample sizes in the growth rate measurements were n=20 for wild-type strains and n=10 for mutant strains. The color and fill pattern of the markers correspond to those in Fig 3. **S3 Fig.** Experimental Evolution of Mutator Strains Under Antibiotics. For all combinations of drugs and strains, the changes in MIC over time are plotted. Each plot overlays data from four replicate series. Dashed lines represent the minimum and maximum MIC values attainable within the constraints of our experimental setup. **S4 Fig.** Reproducibility of the adaptation speed quantification. The MIC doubling rates were estimated by conducting independent experimental evolution trials with varying duration (9 days and 5 days, respectively). Each dot and error bar show the mean and standard deviation of MIC fold change per day, calculated from the data in S3 Fig. The black solid diagonal line means y=x identity line, while the blue line shows linear regression without intercept. The linear regression coefficient and corresponding R2 value were computed for these data points (N=13). **S5 Fig.** The relationship between the mutation rate, calculated by the sum of fixed BPSs and indels, and the MIC doubling rate is shown. Each dot represents an experimental observation from 13 strains across four replicate serial transfer cultures. To prevent overlap of data points, small Gaussian noise (mean = 0, standard deviation = 0.05) was added to the y-coordinates. **S6 Fig.** Mutation Spectra in Resistant Strains. The distribution of substitution patterns and indels calculated by 24 resistant strains (three ancestor mutators, two antibiotics, and four replicates) is presented. Each dot represents the mutation fixation rate during the experimental evolution calculated based on BPSs (closed circle) or indels (open circle). **S7 Fig.** Estimated parameter $\epsilon$ representing the beneficial effect of each mutation. The mean value estimated through 100 bootstrap resampling is represented by the bars, while the error bars represent the standard error. **S8 Fig.** A doubled drug concentration with large β remarkably reduces the growth rate. We defined the net growth rate as μ-γ where μ is the drug-free growth rate and γ is the death rate by drug treatment. $\mu_x$ is the net growth rate with drug concentration $x$, $\mu_{2x}$ is the net growth rate with drug concentration $2x$, and $\mu_{mut2x}$ is the net growth

rate with drug concentration $2x$ for cells with one mutation of beneficial effect $\epsilon$. We assume here a situation where the growth rate and dilution rate are balancing at the drug concentration $x$, i.e., $\mu_x = \ln 100 / 24 \cong 0.19 \left[ h^{-1} \right]$. The black dashed line represents $\mu_x = 0.19$. We used $\varepsilon = 0.9$ and $b_i = 1$ for these simulations. **S9 Fig.** Changes in growth rate during resistance evolution. Growth rates for three strains (WT, S, and LQ) were quantified both before and after experimental evolution. For each selection drug, two independently evolved strains were analyzed. The bars represent the standard deviation of three replicate measurements. **S10 Fig.** Relationship between the parameter *S* and the residual of fitting. The parameter *S*, which describes the saturation effect of mutation accumulation in the form $b_i = b_0 + i\varepsilon S / (i + S)$, is plotted against the residual of the fitting. The modified model incorporating the saturation effect was used to fit the experimental data shown in Fig 3. As observed, the residuals decrease with increasing *S*, suggesting that the saturation effect is negligible.
(DOCX)

**S1 Table.  List of mutations identified in mutation accumulation (MA) experiments.**
(XLSX)

**S2 Table.  Extended summary of MA experiments.**
(XLSX)

**S3 Table.  Colony-size records during MA experiments.**
(XLSX)

**S4 Table.  Common mutations in ancestors which were omitted from the detected mutations in this study.**
(XLSX)

**S5 Table.  Number of generations during the 9-day experimental evolution.**
(XLSX)

**S6 Table.  Number of common mutations among ancestral strains.**
(XLSX)

**S7 Table.  List of mutations in drug-resistant evolved strains.**
(XLSX)

**S8 Table.  Mutational spectrum of wild-type E. coli during drug resistance evolution experiments and drug-free MA experiment in previous studies.**
(XLSX)

## Acknowledgement

We thank Dr. Saburo Tsuru for the fruitful discussions.

## Author contributions

**Conceptualization:** Atsushi Shibai, Minako Izutsu, Chikara Furusawa.

**Funding acquisition:** Atsushi Shibai, Chikara Furusawa.

**Investigation:** Atsushi Shibai, Minako Izutsu, Hazuki Kotani, Chikara Furusawa.

**Methodology:** Atsushi Shibai, Minako Izutsu, Chikara Furusawa.

**Supervision:** Chikara Furusawa.

**Writing – original draft:** Atsushi Shibai, Minako Izutsu, Chikara Furusawa.

**Writing – review & editing:** Atsushi Shibai, Chikara Furusawa.

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
