## [Decision Letter · Decision Letter 0]

4 Sep 2024

Dear Dr Furusawa,

Thank you very much for submitting your Research Article entitled 'Quantitative Analysis of Saturating Relationship between Mutation Rate and Speed of Adaptation under Antibiotic Exposure in Escherichia coli' to PLOS Genetics.

The manuscript was fully evaluated at the editorial level and by independent peer reviewers. All three reviewers agreed that understanding how mutation rates influence antimicrobial resistance is an important topic and that the work presented is interesting. However, the reviewers noted several substantial concerns about the current manuscript. Based on the reviews, we will not be able to accept this version of the manuscript, but we would be willing to review a much-revised version. We cannot, of course, promise publication at that time.

Notably, multiple reviewers voiced reservations about the MutL/DnaQ double mutant and if this combination results in an initial fitness defect independent of mutation rate that would alter its ability to evolve antimicrobial resistance. They note that there may be an inherent defect in the MutL background and that if the other mutants were built upon the MutL strain, this could impact the authors results. Thus, in addition to responding to this concern provide a more complete description of ancestral mutations and how strains are constructed. Additionally, multiple reviewers also commented on the importance of considering how mutator alleles affect not just mutation rate, but also mutation spectrum. The authors may also want to consider the effects of indels as repressor loss of function is a valid source of resistance. Lastly, there is a concern about model fitting, the underlying assumptions of the models used, and how outliers may drive associations. To this end, I also encourage the authors to increase accessibility to the raw data, as providing the data underlying the figures either in a supplementary table or through digital repositories can aid the reviewers in validating the authors’ conclusions.

If you decide to revise the manuscript for further consideration at PLOS Genetics, please aim to resubmit within the next 60 days, unless it will take extra time to address the concerns of the reviewers, in which case we would appreciate an expected resubmission date by email to plosgenetics@plos.org.

If present, accompanying reviewer attachments are included with this email; please notify the journal office if any appear to be missing. They will also be available for download from the link below. You can use this link to log into the system when you are ready to submit a revised version, having first consulted our Submission Checklist .

PLOS has incorporated Similarity Check , powered by iThenticate, into its journal-wide submission system in order to screen submitted content for originality before publication. Each PLOS journal undertakes screening on a proportion of submitted articles. You will be contacted if needed following the screening process.

To resubmit, log into your Editorial Manager account and select the option 'Revise Submission' in the 'Submissions Needing Revision' folder.

We are sorry that we cannot be more positive about your manuscript at this stage. Please do not hesitate to contact us if you have any concerns or questions.

Yours sincerely,

Megan Behringer

Guest Editor

PLOS Genetics

Justin Fay

Section Editor

PLOS Genetics

Reviewer's Responses to Questions

**Comments to the Authors:**

Reviewer #1: The authors’ major conclusion is that there is an optimal mutation rate for the evolution of drug resistances and that a mutation rate higher than the optimum appears to inhibit that evolution. The authors’ simulations of the results lead to the interesting conclusion that high drug sensitivity minimizes the effect of mutation rate on the evolution of drug resistance because fewer cells with beneficial mutation survive. That is, what is important in determining the relationship between mutation rate and speed of evolution is the effective population size under selection. This conclusion will inform our understanding and help to direct further experimentation on the evolution of drug resistances

I do have three major comments. The authors should modify their discussion to incorporate the possibilities that the following factors are also contributing to the results.

1. The main conclusion rests on the observation that the strain with the highest mutation rate failed to have the highest rate of evolution to drug resistances (Figs 3 & 5). But, that strain was a mutLdnaQ double, and it is clear from Fig S1B that the mutLdnaQ strain had a significantly lower growth rate than any other strain even in the absence of drugs. This result suggests that the strain is simply “sick” and that is at least part of the reason it failed to evolve drug resistance. While the authors attribute its growth defect to its high mutation rate, there is another good reason that it might be sick. dnaQ encodes the epsilon subunit of Pol III, the replicative DNA polymerase. Epsilon is part of the core and interacts with the polymerase subunit, alpha, and with theta, the third member of the core. Previous work has shown that interaction of alpha and epsilon is required for fully processive DNA synthesis (Studwell, and O’Donnell, 1990 JBC 265: 1171), and the high rate of deletion formation in dnaQ mutants may be due to these replication defects (Saveson & Lovett, 1997, Genetics 146:457). dnaQ null mutant strains rapidly acquire suppressor mutations that allow DNA replication to recover. The original identified suppressor was in dnaE, which encodes alpha (Lancy et al, 1989, J. Bact 171:5572), but other suppressors are possible, and they might have other phenotypic effects. Clearly, with the loss of MutL, the high mutation rate in the dnaQ mutant is further increased, but the data in this paper do not convincingly show that it is solely the high mutation rate, and not also the loss of other properties due to replication defects, that are responsible for the poor evolution of the mutLdnaQ stain.

2. As shown in Fig. 1 and discussed in lines 137ff; the various strains have different mutational spectra. In particular, the MMR mutants enhance transitions, whereas the presence of the mutT deletion in any strain results in a high rate of AT to CG transversions. This is completely as expected from previous reports. What is striking is that in Fig. 3A&B the strains with the highest evolved drug resistances were mutT mutants (this is assuming I can correctly identify the symbols, some of which are nearly identical). This result suggests that these strains could become drug resistant not just because of their high mutation rate, but also because AT to CG transversions are more likely to give drug resistance than transitions. This possibility could be because the mutT mutants produce a greater proportion of nonsynonymous mutations than the other mutators (Table 1& S2).

3. The three strains with the lowest growth rates are mutL, mutLmutQ, and mutLmuH. This suggests that the mutH and dnaQ alleles were recombined into the mutL single mutant strain that had a growth defect. The sequences of the genetic manipulations are not described, but I tried to confirm this assumption by looking at the mutations labeled “ancestor” of the individual strains in Table S1. I found only an A335V mutant in malP common to all the strains with mutL, including mutTmutL, which did not appear to have a growth defect. So I cannot attribute the low growth rates solely to a pre-existing mutation in the mutL parent before all the other mutant alleles were recombined in, but there does appear to be a growth defect common to all the mutL strains but mutTmutL, suggesting the defect existed at some point in the mutL parent.

Related to this comment: what are the mutations labeled “ancestor”? If, for example, mutS was the starting strain into which mutH and mutT were placed, should not all the “ancestor” mutations in mutS also appear in mutSmutH and mutSmutT? I found only seven mutations common to mut S and mutSmutH, and only four of these were in mutSmutT. This is confusing and the reader should not have to examine the ancestor mutations to understand the genetic manipulations. The authors should describe the constructions more completely and define what the ancestor mutations are.

Specific Comments

57-58: According to the drift-barrier hypothesis articulated in Ref. 2, there is no mechanism for selection to adjust the mutation rate to promote evolution. So this statement is a bit extreme, although the subsequent discussion modifies it.

154: see above about other possible contributions to decreased growth rates

Fig. S1b, Fig 3: It is very difficult to distinguish which strains are indicated by the open circles in these figures. Especially, the symbols for L and HT are nearly identical. The closed circles for SH and LQ also look nearly the same.

179-182: This sentence is difficult to understand

183-185: it would be clearer if the statement excepting the AMK results was first so that the reproducibility statement directly applied to the rest of the experiments.

193: It appears in Fig 3 D & E that the strain with the highest mutation rate, LQ, does have a lower average MIC doubling rate compared to the other mutant strains. The authors might supply some statistics to support this statement

198ff: this reviewer found it easier to follow the mathematical modeling with a table defining the terms, but I still found it difficult. Note that δ is not defined.

220-223: As discussed above, other factors could contribute to growth rate decline. In Fig S1B it is obvious that two strains, L and LH, have close to the same mutation rates as 4 other strains yet have significantly lower growth rates.

290-294: It is not clear why molecular weight of the antibiotic should influence the beneficial effect of a mutation. More likely it is the mechanism of the antibiotic.

351ff: Lukacisinova et al conclude that more sensitive strains evolved resistance faster than more resistant strains. This seems to contradict the conclusions of this paper. Also, Lukacisinova et al argue that using strains without growth defects eliminates the possibility that selection would act on reversing the growth defect instead of antibiotic resistance. This is related to my concern, discussed above. The authors should discuss these differences in approach and conclusions.

372-393: The authors should discuss the results I list in my first comment about dnaQ, viability and suppressors.

Table 1 and S2: A mean of 13.33 synonymous BPS appears 3 times, which seems unusual. Are these correct or was there a miscalculation or typo?

466: AT to CG doesn’t belong in this sentence

Reviewer #2: The manuscript submitted to PLoS Genetics by Shibai et al. reports on the relationship between mutation rate and adaptation rate under antibiotic selection. While the findings are interesting, they are not entirely novel, as previous studies have also shown that a high mutation rate can be costly and does not necessarily lead to better adaptation due to the potential impact on key processes such as replication, transcription, and translation. Although this is an interesting manuscript, it has several weaknesses that need to be addressed before it can be considered for publication.

My major concerns are listed below.

1. When calculating mutation rates, only single nucleotide polymorphisms (SNPs) are considered, which may be problematic. This approach overlooks several loss-of-function mutations, such as indels in a repressor of an efflux operon, that can lead to resistance. Additionally, mutation rates are calculated using only the final time points of evolved cultures, assuming that mutations accumulate monotonically. This assumption might be problematic and not consistent because the rate of adaptation is calculated using continuous phenotype-changing data over time (Figure 2, Figure S2).

2. The rate of adaptation is calculated (Figure S2) by fitting a line to the log(MIC) versus time trajectories. This approach is problematic because adaptation functions can be monotonic, stepwise, or sigmoidal, leading to poor fits in many cases. A better algorithm, such as fitting a log-logistic function, should be used. Alternatively, to be consistent with how mutation rates are calculated, only the net change over 9 days (delta-MIC) can be considered.

3. The assumptions in the population dynamics model are overly simplistic and sometimes problematic. For example, a monotonic increase in the benefit function is assumed as bi=b0+i⋅ϵ. This assumption is difficult to justify in the context of antibiotic resistance evolution. For instance, a single gyrase mutation might be sufficient for bacteria to survive ciprofloxacin, with all subsequent mutations having negligible effects within the experimental range.

4. The authors state, "Moreover, we showed that the relationship between the mutation rate and speed of adaptation is contingent upon the mode of action of the drug involved." However, I am not convinced that this relationship is adequately explained, and I am concerned that the issues highlighted in points #1 and #2 may mislead the authors' conclusions.

5. The claim that growth rates decrease as mutation rates increase is based on the fit in Figure S1, which is unusual and heavily influenced by a single data point.

6. Some of the nomenclature is confusing. For example, what is meant by "mutS mutant"?

Reviewer #3: In this work, the authors have investigated the role of different mutation rates in resistance to five antibiotics. There is a lot I like about this study from an experimental point of view. In particular, I appreciate that the authors performed mutation accumulation to measure the relative mutation rates of the strains. I also like the combination of experiments with parameter estimation.

I was a little disappointed to not see any genetic data from the evolved resistant populations. Particularly, looking at the mutational spectrum from the MA experiments in the different mutators would nicely setup looking at how spectra affects the propensity to evolve different genetic bases for resistance—both in terms of specific mutations within resistance genes, but also the set of genes that acquire mutations. I really think this is a missed opportunity here that would greatly increase the impact of the study.

I would like to have seen some discussion about the epistasis between the mutator alleles themselves. To a first approximation, there are 4 mutation rates here (see Fig3), (1) wild-type (one point), (2) loss of function (LOF) in MutSHL or DnaQ alone (four points), (3) LOF of MutT with or without LOF of MutSHL (six points), and (4) combined LOF of both Mut(SH)L and DnaQ (one point). Changes in MIC were also more or less similar within these clusters.

Joint LOF of DnaQ+Mut(SH)L is clearly an oddity. Given this, it might be an idea to temper the conclusions on the lack of monotonicity, as it’s hard to know whether the evolution of MIC is indeed constrained at higher mutation rates (putatively by the accumulation of deleterious mutations) or because of some synthetically deleterious combination of these alleles. If the authors have this data, it would be nice to compare what the growth rates of the ancestral knockout strains are relative to wild-type, to see if there’s any big defect present already in the LQ strain (or indeed any of the strains). An absence of an initial growth defect would help strengthen the conclusion regarding the non-monotonicity of MIC with mutation rate. The presence of a growth defect would potentially complicate this.

The authors may consider citing a paper from earlier this year by Elgrail et al. (2024, J Evol Biol) on ‘multi-locus mutators’, as it could provide additional support and context for their approach of studying the double knockout strains here. Pubmed link: https://pubmed.ncbi.nlm.nih.gov/38367184/

If I had to find something to comment negatively about, for balance, one area that could be improved was the choice of antibiotics. The authors could alleviate this by mentioning that their study system is for understanding the general principles of resistance evolution, and not specific to particular clinical outcomes. While resistance to each of these can be conferred by spontaneous mutation, mutation isn’t the predominant mechanism of resistance for four out of five of them, outside of laboratory settings. Enzymatic resistance is more relevant in the clinical context for trimethoprim (via acquired dfrA genes), amikacin (aminoglycoside-modifying enzymes), chloramphenicol (chloramphenicol acetyltransferases) and cefixime (beta-lactamases). Ciprofloxacin resistance is, however, acquired by predominantly by mutation. This might not matter for the general readership of PLOS Genetics, but does detract from the applicability to real-world resistance for these particular drugs. To be fair to the authors, they are not explicitly claiming clinical relevance, but there is a tendency toward that thinking implicitly when resistance is studied.

Technical questions:

In general, the experimental design is sound and the analyses performed on the data are appropriate.

1.Why was 34°C used for growth? This is not necessarily problematic, but it differs from others who have largely done experiments at 30°C or 37°C, and this introduces some difficulty when comparing with previous work. Growth temperatures can also affect mutation rates. A brief mention of why might put the readers at ease.

2.Minimal medium with amino acids: Again, no issues with this, but a note about why this medium might be helpful. Particularly, it might help the field to know that rich media such as LB inhibit the effects of some antibiotics e.g. trimethoprim.

3.Insertion sequence (IS)-free E. coli K-12: I don’t think this is a problem, but could the authors please elaborate why they chose to use this interesting strain? I think they have done this because they want to study the pure effects of the mutator alleles. However, IS mobilisation plays an important role in resistance evolution by loss of function, both in laboratory and clinical strains. It would help the reader understanding.

Presentation comments:

Fig1: I like this figure. I assume the pie charts are mutational spectra from the MA experiment, but this isn’t 100% explicit. Might it be helpful in the legend to group the mutations by transition/transversion?

Fig3: I find the colours palette non-intuitive and difficult to distinguish, in particular L/HT (both open points with a blue/green colour), and S/QT (two shades of yellow). The open-yellow Q points are also very hard to see. I can’t make out why some treatments were assigned open points.

I would suggest two things. First, use shapes in addition to colours. For instance, black squares for WT, coloured circles for the single-locus mutators (H,L,S,Q,T) and then coloured triangles for the two-locus mutators (LH,SH, LQ, etc…).

Second, group the labels according to the ‘clusters’ they form on the x-axis. If you group the labels in this way, then you only need to take care that the colours are very distinct within these. There are some nice options for six-colour palettes here: https://coolors.co/palettes/popular/6%20colors

FigS1: Repeat the colour scale legend here as it’s hard for the reader to make them flip between documents to read the figures, especially as it’s the same complicated colour scale as in Fig3.

I hope that the authors find these comments to be useful.

Danna Gifford

**Have all data underlying the figures and results presented in the manuscript been provided?**

Reviewer #1: Yes

Reviewer #2: Yes

Reviewer #3: **No: ** Raw data for figures was *not* available during review, either as part of the supplementary excel file or in the Data Availability statement

PLOS authors have the option to publish the peer review history of their article (what does this mean? ). If published, this will include your full peer review and any attached files.

**Do you want your identity to be public for this peer review?** For information about this choice, including consent withdrawal, please see our Privacy Policy .

Reviewer #1: No

Reviewer #2: No

Reviewer #3: **Yes: ** Danna Gifford

---

## [Decision Letter · Decision Letter 1]

10 Jan 2025

PGENETICS-D-24-00811R1

Quantitative Analysis of Saturating Relationship between Mutation Rate and Speed of Adaptation under Antibiotic Exposure in Escherichia coli

PLOS Genetics

Dear Dr. Furusawa,

Thank you for submitting your manuscript to PLOS Genetics. After careful consideration, we feel that it has merit but does not fully meet PLOS Genetics's publication criteria as it currently stands. Therefore, we invite you to submit a revised version of the manuscript that addresses the remaining points raised during the review process.

We appreciate the careful revisions and thoughtful responses provided in this recent submission. However, there are still a couple of issues raised by the reviewers that merit discussion. In addition to the comments below, we also appreciate the newly-added indel data in the mutation rate and spectra results, and believe it elevates the impact of the manuscript. However, these new results deserve further discussion and contextualization. In particular, the increased indel frequency is primarily observed in the lower tier mutators and the LQ outlier, while indels are considerably rare in the higher tier mutators. As indels are expected to result in larger effects on average compared to base pair substitutions, and resistance mutations often arise in essential genes where disruption by indels can be lethal, it is important to consider the implications of the differing mutation spectra on your results.

Please submit your revised manuscript within 30 days Feb 09 2025 11:59PM. If you will need more time than this to complete your revisions, please reply to this message or contact the journal office at plosgenetics@plos.org. Please include the following items when submitting your revised manuscript:

We look forward to receiving your revised manuscript.

Kind regards,

Megan Behringer

Guest Editor

PLOS Genetics

Justin Fay

Section Editor

PLOS Genetics

Aimée Dudley

Editor-in-Chief

PLOS Genetics

Anne Goriely

Editor-in-Chief

PLOS Genetics

**Journal Requirements:**

1) Please upload all main figures as separate Figure files in .tif or .eps format. For more information about how to convert and format your figure files please see our guidelines:

2) Please ensure that the funders and grant numbers match between the Financial Disclosure field and the Funding Information tab in your submission form. Note that the funders must be provided in the same order in both places as well.

**Reviewers' comments:**

Reviewer's Responses to Questions

**Comments to the Authors:**

Reviewer #1: The authors have satisfactorily addressed most of my specific comments but my overall opinion of this paper is not changed significantly by the revisions. While the results are interesting, the authors’ interpretations rely heavily on the behavior of one strain, the mutL dnaQ double mutant (LQ). Indeed, the relationship between mutation rates and the increases in MIC of the strains in Figure 3 can be interpreted in every case as linear with one clear outlier, LQ. Likewise, the relationship between mutation rates and growth rates of the strains in Fig. S2B can be interpreted as a straight line, no effect, with LQ again being a clear outlier and L and LH also lying below the line. All three original reviewers brought this up; reviewer #3 was most articulate, pointing out the possibility of “synthetically deleterious combination of these alleles”. The results in Fig. S2B verify this possibility. The authors have addressed this interpretation in the text, justifying their modeling by with the assumption that the effects of any detrimental factors are proportional to the effects of the observed mutation rates. But this assumption has no evidential support. Factors that reduce growth rates may directly or indirectly also reduce mutation rates and/or the rates of MIC increase.

These considerations cloud the interpretation of the results. I feel that most readers, if they read carefully, will come to the same conclusion.

One new comment. The authors repeatedly attribute the growth defects of the LQ strain to hyperactivation of mismatch repair (MMR) and reference Stefan et al, Ref. 35, for this possibility. But, they have misinterpreted Ref. 35. In Stefan et al, MMR is not inferred to affect the growth rate, but to act to keep the mutation rate low when cell growth is slow due to the silencing of dnaQ. They conclude: “the ε subunit primarily affects the processivity of DNA polymerase III” and that is what causes the growth defects.

The authors also reference a paper by Iyer et al, 2020, showing that overexpression of MutS causes double-strand breaks and inhibits cell division. However, these effects were with MutS dramatically overinduced. At lower levels (10x normal) MutS overexpression had no growth effect and, in fact, elevated some mutation rates, which effect was suppressed by overexpression of MutL. Also, as far as this reviewer can find, this paper did not appear in a peer-reviewed journal.

Reviewer #2: The authors addressed almost all my concerns, except for their linear model used for benefit change (bi = b0 + i⋅ϵ). However, this does not impact my recommendation for publication, as such differences in scientific opinions are both expected and justified. Therefore, I recommend the manuscript for publication.

Reviewer #3: I thank the authors for their high attention to detail and cleanly presented response to comments.

Regarding the new genetic information, I am surprised the authors did not make a little more about the information. Specifically, the fact that 'expected' mutations in the canonical resistance target, gyrA, were found in the ciprofloxacin populations, and ampC mutations in the cefotaxime populations, I think warrants mention.

Otherwise I am satisfied with the changes the authors have made.

Danna

**Have all data underlying the figures and results presented in the manuscript been provided?**

Reviewer #1: Yes

Reviewer #2: Yes

Reviewer #3: Yes

PLOS authors have the option to publish the peer review history of their article (what does this mean? ). If published, this will include your full peer review and any attached files.

**Do you want your identity to be public for this peer review?**  For information about this choice, including consent withdrawal, please see our Privacy Policy .

Reviewer #1: No

Reviewer #2: No

Reviewer #3: **Yes: ** Danna Gifford

**Figure resubmission:**
---

## [Editor Report · Decision Letter 2]

17 Feb 2025

Dear Dr Furusawa,

We are pleased to inform you that your manuscript entitled "Quantitative Analysis of Relationship between Mutation Rate and Speed of Adaptation under Antibiotic Exposure in Escherichia coli" has been editorially accepted for publication in PLOS Genetics. Congratulations!

Yours sincerely,

Megan Behringer

Guest Editor

PLOS Genetics

Justin Fay

Section Editor

PLOS Genetics

Aimée Dudley

Editor-in-Chief

PLOS Genetics

Anne Goriely

Editor-in-Chief

PLOS Genetics

Comments from the reviewers (if applicable):

**Data Deposition**

http://datadryad.org/submit?journalID=pgenetics&manu=PGENETICS-D-24-00811R2

**Press Queries**

---

## [Editor Report · Acceptance letter]

PGENETICS-D-24-00811R2

Quantitative Analysis of Relationship between Mutation Rate and Speed of Adaptation under Antibiotic Exposure in Escherichia coli

Dear Dr Furusawa,

We are pleased to inform you that your manuscript entitled "Quantitative Analysis of Relationship between Mutation Rate and Speed of Adaptation under Antibiotic Exposure in Escherichia coli" has been formally accepted for publication in PLOS Genetics! Your manuscript is now with our production department and you will be notified of the publication date in due course.

With kind regards,

Anita Estes

PLOS Genetics

On behalf of:
